# High-Throughput Bacteriophage Testing with Potency Determination: Validation of an Automated Pipetting and Phage Drop-Off Method

**DOI:** 10.3390/biomedicines12020466

**Published:** 2024-02-19

**Authors:** Nicolas Dufour, Raphaëlle Delattre, Laurent Debarbieux

**Affiliations:** 1Réanimation Médico-Chirurgicale, Hôpital NOVO—Site de Pontoise, 95300 Pontoise, France; 2Institut Pasteur, Université Paris Cité, CNRS UMR6047, Bacteriophage Bacterium Host, 75015 Paris, Francelaurent.debarbieux@pasteur.fr (L.D.); 3IAME, Université de Paris, INSERM U1137, Université Sorbonne Paris Nord, 75018 Paris, France; 4Réanimation, Centre Hospitalier de Digne-les-Bains, 04000 Digne-les-Bains, France

**Keywords:** phage therapy, phagogram, automation, spot test, variability, reproducibility

## Abstract

The development of bacteriophages (phages) as active pharmaceutical ingredients for the treatment of patients is on its way and regulatory agencies are calling for reliable methods to assess phage potency. As the number of phage banks is increasing, so is the number of phages that need to be tested to identify therapeutic candidates. Currently, assessment of phage potency on a semi-solid medium to observe plaque-forming units is unavoidable and proves to be labor intensive when considering dozens of phage candidates. Here, we present a method based on automated pipetting and phage drop-off performed by a liquid-handling robot, allowing high-throughput testing and phage potency determination (based on phage titer and efficiency of plaquing). Ten phages were tested, individually and assembled into one cocktail, against 126 *Escherichia coli* strains. This automated method was compared to the reference one (manual assay) and validated in terms of reproducibility and concordance (ratio of results according to the Bland and Altman method: 0.99; Lin’s concordance correlation coefficient: 0.86). We found that coefficients of variation were lower with automated pipetting (mean CV: 13.3% vs. 24.5%). Beyond speeding up the process of phage screening, this method could be used to standardize phage potency evaluation.

## 1. Introduction

The development of bacteriophages (phages) as active pharmaceutical ingredients for the treatment of patients infected with antibiotic-resistant or difficult-to-treat bacteria is now on the agenda of health regulatory authorities such as the European Directorate for the Quality of Medicines and healthcare (EDQM) and the Food and Drug Administration (FDA) [1,2].

Given the specificity of phages to infect a limited number of bacterial isolates within a bacterial species, the development of phage therapy for each patient necessitates a large collection of phages [3]. Then, the next step is invariably to test these numerous phages to select those that exhibit the best activity, which corresponds to the highest number of produced plaque-forming units (PFUs) and to the longest period preventing the growth of phage-resistant bacteria. These characteristics help define the pharmaceutical potency of a phage.

During the past decade, several laboratories and a few institutions have engaged in building large phage collections [4,5,6,7]. As the number of phages to be tested manually is increasing, there is a need for an automated process, which could be more time efficient, and less labor intensive. The benefit of test automation is undeniable when considering standardization and reproducibility. In the field of clinical laboratories (from biochemistry to microbiology, through hemostasis or toxicology), accreditation is now mandatory in most countries [8]. Without accreditation, results produced from any analysis could be regarded as meaningless (unreliable results). In most cases, accredited laboratories must comply with ISO 15,189 and 17,025 standards, testifying for reproducibility and repeatability of measures, as well as minimization of human-dependent errors. Considering the development of phage therapy and its future entry into medical microbiology laboratories, it appears necessary to consider phage-related diagnosis tests as a target of automation.

Here, we present and discuss an automated technique using a liquid-handling robot, allowing rapid high-throughput susceptibility testing. The aims of this study were: (i) to validate an automated method of pipetting and spotting to perform phage susceptibility and potency testing: we carried out a comparison to a reference (manual) method (one bacterial strain, one phage), and (ii) to test the feasibility and limits of the automated method on a large series of measures (10 phages and an assembly of these into a cocktail against 126 *Escherichia coli* strains).

## 2. Materials and Methods

### 2.1. The Manual Drop-Off Method

The reference method routinely used in our laboratory, sometimes referred to as “direct spot test” [9] belongs to the group of plaque-based assays. As the classical double-layer agar method, it allows the detection of individual PFUs and the determination of the titer of a phage suspension. This direct spot test method consists of spotting drops (4–10 µL) of serial 10-fold dilutions (performed in a 96-well plate) of a phage stock, on the surface of an agar plate (LB-Lennox) previously inoculated with the bacteria to be tested.

Inoculation is obtained by inundation (1 mL per round agar plate) with a fresh exponentially growing (OD_600_ 0.3–0.4) liquid culture (LB-Lennox, 37 °C, 200 rpm) of the strain to be tested to cover the entire surface of the agar plate, to obtain a thin and homogenous bacterial lawn, as it is intended when performing antibiotic susceptibility testing [10]. The excess of liquid is then removed by pipetting and the plate is kept open under a safety cabinet for a few minutes to allow the surface to dry.

Next, phage dilutions are spotted, in triplicate, on the surface of the bacterial lawn, using an 8-channel pipette. Particular attention is required regarding the type of tips, which must be highly hydrophobic to allow, once ejection has taken place, the total detachment of the drop formed at the extremity of the tip, triggered by gentle contact of this drop with the agar surface. Once the drops of the phage spots are soaked up, the plates are incubated at 37 °C, usually overnight. After incubation, the number of individual plaques is counted and the phage titer is calculated.

Based on a few independent assays to titrate virulent phages infecting either *Pseudomonas aeruginosa* or *Escherichia coli*, this method proved to be as accurate as the classical double-layer agar assay. Aside from being notably less labor intensive, it also avoids brief thermal stress to bacteria and phages imposed by using the molten soft agar (50–55 °C).

### 2.2. The Automated Drop-Off Method

#### 2.2.1. Robot, Tips, and General Setup

In 2014, we refurbished a liquid-handling robot manufactured by Tecan (model Genesis, Tecan, Switzerland) equipped with a single arm carrying four pipetting channels, each connected to 1000 µL syringes. The robot was controlled by proprietary software (GENESIS Instrument Software V4.21), on which the run programming was performed. Briefly, the robot’s arm provided displacements of the four pipetting channels in the X and Y axes, which could then independently move along the Z axis (up and down) and equip themselves automatically with tips. Tips containing graphite were used, allowing liquid detection using conductivity changes between air and liquid (Tecan, 200 µL disposable tips, “Non-filtered, Pure”). The robot’s working area was equipped with a homemade Plexiglas^®^ flat tray to handle up to 10 square agar plates (see below).

All plates used during the assay (the 96-well plate containing the phages and the square agar plates containing the bacterial lawn) were formerly characterized in terms of size (length, width, height) and positioned using the instrument software. For the robot, this ensured a precise location of each well and drop-off areas on the agar surface in terms of X, Y, and Z axis coordinates. The program, the liquid class definition, and the settings of used labware are available in the Appendix A: configuration files).

#### 2.2.2. Agar Plates

Sterile square Petri plates (15.8 × 120 × 120 mm, Gosselin, France) were placed on a horizontal plane and filled precisely with 50 mL of LB-Lennox agar medium to guarantee a constant thickness and therefore a constant height in the Z axis. As described for the manual method (see Section 2.1), a liquid culture (4 mL) of the tested bacteria was poured on each plate before use (see below). The plate was then tilted to remove the excess of liquid by gentle pipetting. Special attention was given to the dryness of the agar plates as this parameter is critical to avoid drops spreading and coalescence. This was controlled by placing opened agar plates in a biological safety cabinet for 15–30 min.

Before being placed in the robot’s working area, each plate was labeled on the top left corner with the name of the tested strain for identification and orientation purposes (subsequent result readings).

#### 2.2.3. Bacteriophages

Ten *E. coli* phages from our lab collection (the *Antonina Guelin* collection [11] and unpublished ones) and an assembly of these 10 phages (subsequently designated as the cocktail) were used for this study (phages listed in Appendix A). Phage stocks (>10^8^ PFU/mL) corresponding to filter-sterilized lysates (in LB-Lennox) were distributed in a 96-deep well polystyrene plate (1.2 mL, TreffLab, Switzerland). Between runs, this plate was stored at 4 °C. When the 96-well plate was positioned in a landscape view, the first 11 rows (vertical) corresponded to the 10 single phages and finally the cocktail. Lines 1 to 8 (horizontal) corresponded to the ten-fold serial dilutions (from non-diluted to 10^−7^).

#### 2.2.4. Strains

A total of 126 *E. coli* strains were tested for their susceptibility to the above-mentioned phages (strains listed in Appendix A). Among these 126 strains, 10 were positive controls (i.e., strains that are the hosts of the 10 phages tested) while the others were randomly picked from a large collection of *E. coli* isolates of various origins hosted by our laboratory and the IAME laboratory (Infection, Antimicrobials, Modelling and Evolution; INSERM unit, UMR 1137, Paris).

#### 2.2.5. Step-by-Step Description of Robot Processes during a Run

After launching the program, the user entered the number (*n*) of bacterial strains to be tested per run (i.e., the number of agar plates, *n* = 1 to 10) and the volume of the drop (*v*, 5 µL). The sequence started with the arm moving the pipetting channels to pick up four tips from the tips box and then moving above the first four wells (A1 to A4) of the 96-well plate containing phages, to aspirate a total volume of (3*vn* + 10) µL per tip. Then, the robot’s arm moved to sequentially distribute four drops of 5 µL each on the surface of the *n* agar plates, each overlaid with the bacterial lawn. This was performed in triplicate as illustrated in Appendix A (the video illustrates the distribution of two dilutions of phage, in triplicate, on one agar plate).

At the end of the distribution, tips were ejected and a new cycle started with the next four wells (B1 to B4) and so forth. The software records indicate that a full run (testing 4 phages at 8 different concentrations in triplicate on 10 bacterial strains, i.e., 96 spots per agar plate and 960 spots in total) lasted 24 min. Three consecutive runs were thus needed to distribute all dilutions of the 10 phages and the cocktail on 10 bacterial strains.

After completion of the run, plates were incubated for 8–12 h at 37 °C.

### 2.3. Validation of Concordance between Manual and Automated Methods

To compare the results obtained with the manual and the automated methods, the phage 536_P1 (*Myoviridae*, 149.4 kbp) and its isolation strain (*E. coli* 536) were used [12]. The same lot of phage stock, diluent (LB-Lennox) and agar plates were used. Four different dilutions were prepared from the phage stock: undiluted (tube A), 10^−1^ (tube B), 10^−2^ (tube C) and 10^−3^ (tube D).

Each tube (A to D) was then subjected to titration in triplicate, by using the manual and the automated method as described above (Section 2.1 and Section 2.2). Four independent replicates (agar plate and dilution) were performed in each case.

### 2.4. Plates Reading, Data Analysis and Statistics

After the incubation period, agar plates were analyzed as follows, whatever the method considered (manual or automated pipetting): only phages able to form individual plaques were considered active on the considered strain. Phages leading to partial or entire bacterial lawn clearance without forming individual plaques at higher dilutions were removed from the analysis as titers could not be calculated. Considering the three replicates, a mean concentration (knowing the volume of the drop and the dilution factor), its standard deviation and its coefficient of variation (standard deviation divided by the mean) were calculated.

We only focused on intra-observer variability as only one user performed both the manual pipetting and readings of the plates (i.e., PFU count).

Results from both methods were analyzed in terms of fidelity (i.e., ability to reproduce the same result when repeated) and accuracy (i.e., ability to indicate the truth, compared to a reference method) using methodologies dedicated to such analysis [13] like the Bland and Altman approach [14] and Lin’s concordance correlation coefficient [15]. Bland and Altman representation and calculation were performed using the ratio of titers obtained by repeated measurements (*n* = 16, automated method divided by manual method) after a Log10 transformation of the values.

Statistical analyses were performed using GraphPad Prism version 7.03 (GraphPad Software, CA, USA). The normal distribution of all variables was checked using the Kolmogorov–Smirnov test. Statistical tests (Student’s t-test or the Mann–Whitney test) were chosen accordingly. A P value less than 0.05 was considered statistically significant. Lin’s concordance correlation coefficient was calculated using the Real Statistics Resource Pack software (Release 8.9.1, copyright 2013–2023) created by Charles Zaiontz (www.real-statistics.com) on Microsoft Excel.

## 3. Results

### 3.1. Comparison of the Automated to Manual Pipetting and Drop-Off Methods Using Phage 536_P1

Using the same source material (phage, bacteria, media), we performed, four times independently, the titration of phage 536_P1 on strain 536 in triplicate using both the manual and automated pipetting and drop-off methods.

The difference in phage titers observed between the two methods was of negligible meaning (Table 1 and Figure 1). When pairwise comparisons were performed on the mean of values obtained from the four replicates, the absolute difference between the two methods was 0.020 Log for tube A (1.03 × 10^9^–1.08 × 10^9^), 0.042 Log for tube B (1.18 × 10^8^–1.30 × 10^8^), 0.075 Log for tube C (1.05 × 10^7^–1.25 × 10^7^) and 0.204 Log for tube D (1.65 × 10^6^–1.03 × 10^6^). Taken together, the mean absolute difference for all measures was 0.085 Log.

The highest variability (33.93%) expressed by the coefficient of variation (CV) was observed with the manual method on tube A (Table 1). While this was expected as tube A contained the highest phage titer (likely to amplify any pipetting inconsistency), we found that the CV obtained with the automated method for the same tube was lower (20.94%). The same trend was observed for the three other tubes (B, C, D); and overall, the variability was significantly lower when the titration was performed by the robot as compared with the manual titration (CV = 13.30 vs. 24.48%, respectively, *p* = 0.04).

The Bland and Altman analysis using a ratio vs. average method showed a very low bias of 0.99 (95% confidence interval: 0.95–1.05), with a stable concordance of the two methods across the interval of tested phage titers (roughly 10^6^ to 10^9^ PFU/mL) as shown in Figure 2. In this analysis, the closer the ratio (of results from one method to the other) is to 1, the more these methods give identical results. Finally, Lin’s concordance correlation coefficient was 0.86 (95% confidence interval: 0.66–0.95), indicating a good agreement between the two methods. A quasi-equivalence was then demonstrated between the manual and automated drop-off methods.

### 3.2. Evaluation of the Performance of the Automated Method on a Large Series of Tests

Next, we used the robot to spot serial dilutions of 11 phage suspensions (10 individual phages and a cocktail of) on a set of 126 *E. coli* strains that included the 10 original hosts of the phages (positive controls). The latter were all susceptible to their respective phages.

A total of 33,264 spots were performed by the robot: 126 strains × 11 phages (10 singles phages and 1 cocktail) × 8 concentrations × 3 replicates. We used 378 agar plates (3 plates per strain), which corresponds to 38 runs (1 run = 10 plates). Overall, the duration of these tests lasted about 16 h (38 × 24 min), which were split into three consecutive working days.

Within the limits presented hereinafter, drops of phage suspensions were evenly distributed on the surface of the agar plate (Appendix A). Nonetheless, the most frequently observed irregularity was the merging of two adjacent spots (Figure 3, Panel B), which occurred in 1.97% of the total spot (i.e., approximately 1 coalescence for 50 spots or 2 coalescences per plate on average). Opportunely, due to the proceeding of the run, these events were only observed within replicates and never between drops of different phages or different dilutions. The consequence of such a merge had a weak impact as PFUs were still separated from one spot to the other (despite merging of the drop) or because the PFUs count from the merged spots could be divided by 2 and remained still close to the third replicate.

The second irregularity linked to the automation was a “no drop-off” error when a spot was missed (Figure 3, Panel A): it occurred in 0.93% (i.e., approximately 1 spot missing out of 100, 1 occurrence per plate). This was mainly observed when a drop stayed attached to the corresponding tip and less frequently to default in liquid aspiration. From a qualitative point of view, the consequence was limited thanks to the existence of replicates. However, having only two replicates instead of three (with a possible double dose in phage drop-off on one spot for some) has indisputably elevated the CV in these rare events (Figure 4).

From a total of 151 positive phage–bacteria interactions, for which individual PFUs were identified, four of them were excluded from the analysis as the number of individual plaques in the highest dilution was still above 50 and precluded a reliable count. The remaining 147 positive interactions corresponding to 441 spots (147 × 3) were analyzed. The concentration range was 5 × 10^3^ to 7 × 10^10^ PFU/mL. The median coefficient of variation for all measures was 16.9% (IQ_25–75%_ range: 10.6–27.5%, Figure 4).

When focusing on the precision (fidelity) of the measures, we found that an acceptable value (defined here as a CV lower than 15%) was observed when the phage titer was assessed by counting at least 15 PFUs per spot. In other words, the lowest variation in repeated measures was obtained when the number of counted events (PFU per spot) was higher than 15 (Table 2).

In this report, we did not exploit the data beyond those needed for the critical analysis of the automated method. However, it should be noted that we could: (i) calculate the efficiency of plaquing for each phage forming individual plaques on tested strains, (ii) detect the growth of phage-resistant bacteria within clear spot areas and (iii) observe different plaque morphologies with cocktail as shown in Figure 3, Panel C.

## 4. Discussion

We demonstrated here that a liquid-handling robot was suitable to perform direct spot tests of serial phage dilutions for the evaluation of phage potency in semi-solid medium (i.e., presence or absence of a lytic activity, titration, and efficiency of plaquing when lysis is observed). In particular, we found that the variability of measures obtained with the robot was lower, as compared with the manual method. The challenge represented by dropping off a low volume of liquid on the surface of an agar plate required several steps of optimization but was overall well achieved and proved to be stable over several independent runs.

Within a triplicate, the observed coefficient of variation for our measures (titers) was nonetheless slightly higher than the one usually reported in the literature with an automated pipetting system (in terms of volume accuracy), which is usually lower than 10% [16,17]. Nonetheless, one has to keep in mind that our analysis of variability concerned the final result of two operations: the pipetting itself (aspiration, serial ejection) and the drop-off on the agar surface (that must be complete), the latter being not evaluated in studies focused on pipetting accuracy.

Our work has several limits. First, we only assessed intra-observer variations, as the purpose of this study was only to investigate the technique by itself and not the whole process in multiple hands (i.e., the inter-observer variations) [18]. By definition, operations performed by an automated system display no or very weak variations. If present, these variations are within the limit of the technical specifications of the process and not subject to the influence of human interventions. Second, phage potency testing might not be limited to assays performed in semi-solid (agar) medium. Interactions between phages and bacteria are different in liquid and in semi-solid medium. Indeed, phenotypic traits in bacteria are highly different according to their planktonic or sessile lifestyle [19]. Changes displayed are prone to introduce many differences in phage–bacteria interaction through, for example, variations in expression of receptor binding proteins, capsule, metabolic state (linked to available nutrients), and rate of cell division [20]. Consequently, differences in phage susceptibility testing results between semi-solid or liquid medium can sometimes be observed. In most cases, assays in liquid medium offer the possibility to follow bacterial lysis over time by recording the optical density at regular intervals and provide data on the ability of the phage population to hold onto a low bacterial density without the apparition of resistant clones over time [21]. Fortunately, studies of the interactions between phages and bacteria in a liquid medium could also be implemented with a liquid-handling robot. Third, we observed that at least 15 PFUs should be numerated to decrease the measure variability in the evaluated conditions. This target was difficult to achieve with phages making large plaques as less than ten individual plaques could be enough to merge altogether into a complete clearance area. For those situations, either an early reading of the plate or a conventional method (like the double-layer agar method, that provides a larger surface of analysis) could help circumvent this limitation.

To assess phage potency, the visualization of individual PFUs is of critical importance and the methods presented here (manual or automated) allow us to reach this objective. Beyond allowing the precise determination of the viral titer of a solution, this point is technically important to prevent misclassification of phage activity that could be linked to mechanisms of lysis that do not rely on a viral cycle but, as an example, on an abortive infection [22] or lysis from without [23].

In these cases, a total or partial erasure of the bacterial lawn may be observed at high phage concentration but without individualized PFUs at lower dilutions. When present, these observations may not be related to a viral cycle, and phages that demonstrate such a pattern on a given strain should not be considered active, according to the classical definition (that is for a virus, the ability to replicate into the host and produce progeny).

Assessing precise titers also allows, as in our method, to calculate the EOP (efficiency of plaquing) of a phage towards all susceptible strains. EOP is the titer of a phage against a specific strain, as compared to its titer on its reference host strain (titrated in parallel) [24]:$$EOP=\frac{Titer\;of\;phage\;X\;on\;the\;tested\;strain\;\left(PFU/mL\right)}{Titer\;of\;phage\;X\;on\;its\;reference\;host\;strain\;\left(PFU/mL\right)}$$

On this basis, phages can be classified in terms of potency, using the following commonly accepted thresholds: highly virulent (0.1 < EOP > 1.00), moderately virulent (0.001 < EOP < 0.099), and weakly or avirulent (EOP < 0.001) [25,26,27].

However, the simplest method like the classical spot test (or spot assay), often reported in the literature [24], still has the advantage of the rapidity and can be used for large and quick screenings. As such, the spot assay (one drop of phage without serial dilutions) may provide only qualitative results with a lower specificity. Thus, the inherent weakness of this approach [24,28], should be kept in mind, especially the risk of false positives linked to the above-mentioned mechanisms, as well as the impossibility to calculate an EOP value.

Automation of clinical microbiology laboratories is continuously growing and provides many benefits to biologists, clinicians and ultimately patients. Among them, automation improves reproducibility, traceability, and time to test result, while decreasing human related errors, repetitive tasks, and possibly long-term financial costs [29]. We can anticipate that, even if expensive and requiring dedicated training, automated phage titration techniques as presented here could be routinely used by clinical laboratories. Nonetheless, a complete automation process should not be considered as such if it does not include the last part of the process, which is plate reading and PFU count. Fortunately, automation of these tasks is already under development, which will certainly contribute to improving phage potency assessment [30,31,32].

## Figures and Tables

**Figure 1 biomedicines-12-00466-f001:**
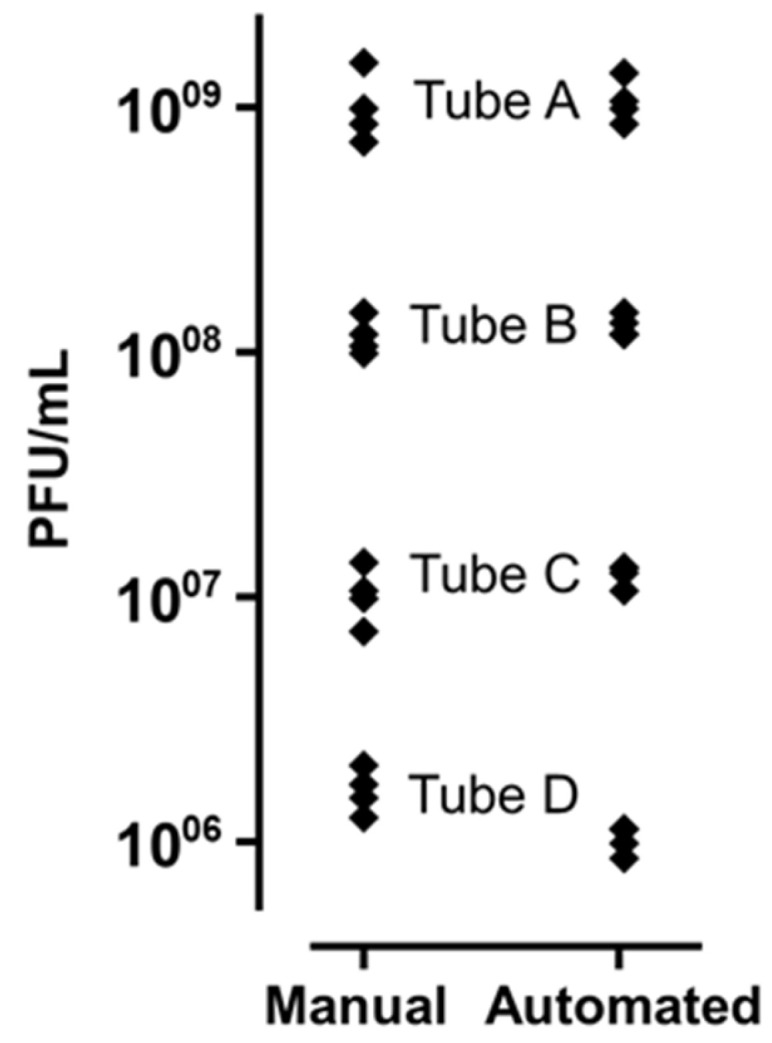
Phage 536_P1 titers on strain 536, obtained from the manual and automated drop-off methods. The results of four independent replicates are shown, at four different concentrations (Tube A to D).

**Figure 2 biomedicines-12-00466-f002:**
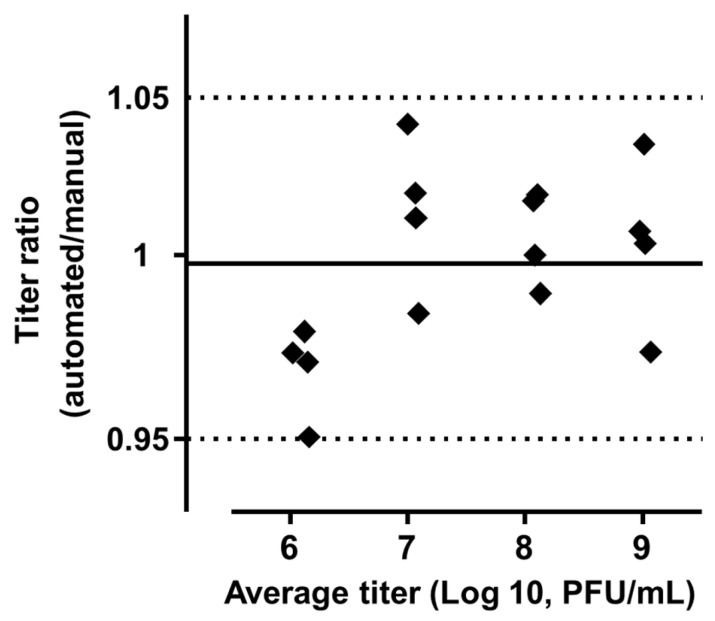
Bland and Altman representation of the ratio of phage titers according to the average phage titer. A total of 16 couples of measures is shown. The solid line represents the bias and the dashed lines represent the 95% agreement interval between the two methods.

**Figure 3 biomedicines-12-00466-f003:**
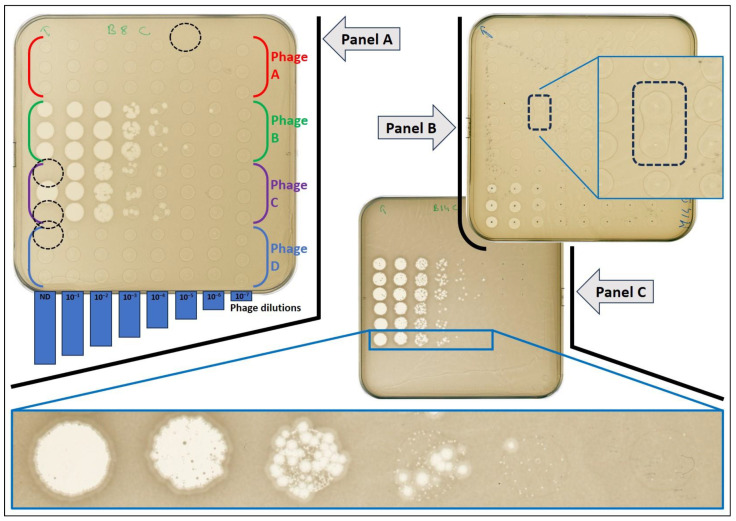
Representative images of agar plates obtained with the automated method. **Panel A:** layout of phage spots on the bacterial lawn (four different phages, vertical axe, A to D), at a decreasing concentration (horizontal axe, from left to right, ten-fold serial dilution, non-diluted to 10^−7^), in triplicate. Dotted circles indicate the absence of a drop at the expected location. **Panel B**: example of drop coalescence (merging). **Panel C**: example of the visualization of two plaque morphologies when testing the cocktail, corresponding to two active phages.

**Figure 4 biomedicines-12-00466-f004:**
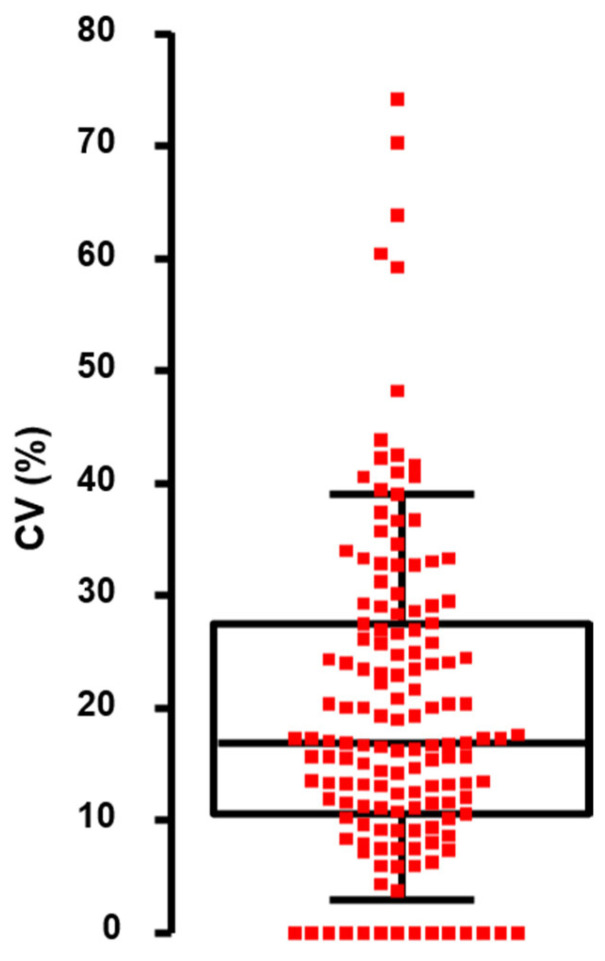
Distribution of the coefficient of variation (CV) of the phage titers obtained from 147 positive interactions. Each dot is the CV of one titration performed in triplicate. A CV of 0 means that no variation was observed within the triplicate (equal number of PFUs on each replicate). The box plot indicates median and interquartile ranges (25–75%); whiskers represent the 10th and 90th percentiles.

**Table 1 biomedicines-12-00466-t001:** Phage 536_P1 titers on strain 536, obtained from the manual and automated drop-off methods, at four different concentrations.

Dilutions	Manual Method (Reference One)				Automated Method (Evaluated One)			
	A (ND)	B (10^−1^)	C (10^−2^)	D (10^−3^)	A (ND)	B (10^−1^)	C (10^−2^)	D (10^−3^)
Rep. #1	1.00 × 10^9^	1.00 × 10^8^	7.33 × 10^6^	1.53 × 10^6^	1.07 × 10^9^	1.33 × 10^8^	1.33 × 10^7^	1.13 × 10^6^
Rep. #2	7.33 × 10^8^	1.20 × 10^8^	1.40 × 10^7^	1.73 × 10^6^	8.67 × 10^8^	1.20 × 10^8^	1.07 × 10^7^	1.13 × 10^6^
Rep. #3	1.53 × 10^9^	1.07 × 10^8^	1.07 × 10^7^	2.07 × 10^6^	1.40 × 10^9^	1.47 × 10^8^	1.27 × 10^7^	1.00 × 10^6^
Rep. #4	8.67 × 10^8^	1.47 × 10^8^	1.00 × 10^7^	1.27 × 10^6^	1.00 × 10^9^	1.20 × 10^8^	1.33 × 10^7^	8.67 × 10^5^
**Mean of Rep.**	**1.03 × 10^9^**	**1.18 × 10^8^**	**1.05 × 10^7^**	**1.65 × 10^6^**	**1.08 × 10^9^**	**1.30 × 10^8^**	**1.25 × 10^7^**	**1.03 × 10^6^**
SD	3.51 × 10^8^	2.06 × 10^7^	2.74 × 10^6^	3.37 × 10^5^	2.27 × 10^8^	1.28 × 10^7^	1.26 × 10^6^	1.28 × 10^5^
**CV (%)**	**33.93**	**17.44**	**26.11**	**20.44**	**20.94**	**9.82**	**10.10**	**12.35**
**Mean CV (%)**	**24.48 (±7.25)**				**13.30 (±5.21)**			

ND: non-diluted, Rep.: independent replicate, SD: standard deviation, and CV: coefficient of variation.

**Table 2 biomedicines-12-00466-t002:** Phage titration using the automated method: mean coefficient of variation (CV) according to the number of plaques counted per spot within a triplicate (147 titrations analyzed).

Number of PFUs Per Spot (Mean of the 3 Replicates)	Number of Independent Triplicates Analyzed	Mean CV (%)
]0–5]	6	37.1%
]5–10]	26	29.3%
]10–15]	37	25.5%
]15–20]	40	14.1%
]20–25]	19	14.1%
]25–30]	12	9.6%
]30–50]	7	7.9%

## Data Availability

The raw data supporting the conclusions of this article will be made available by the authors on request.

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
