# Peer review of "High-Throughput Bacteriophage Testing with Potency Determination: Validation of an Automated Pipetting and Phage Drop-Off Method"

_biomedicines, 2024, doi:10.3390/biomedicines12020466_

Round 1
Reviewer 1 Report
Comments and Suggestions for Authors
This work developed a bacteriophages potency testing method based on an automated pipetting and phages drop-off performed by a liquid-handling robot, which allows high throughput testing. The comparison with manual assay shows good reproducibility and concordance, lower coefficients of variation. This is an interesting and useful work that will benefit the field. I would suggest to accept it for publication in its current form.
Author Response
R: We thank the reviewer for the positive comments.
Reviewer 2 Report
Comments and Suggestions for Authors
The manuscript of Dufour et al. validates a semi-automated method to assess bacteriophages potency. The method described shows good reproducibility and will be important for speeding up the process of bacteriophage screening, in order to identify good therapeutic candidates for each specific bacterial infection.
Revisions:
1) In the manuscript the term “semi-automated” and “automated” is used. However, these terms have different meanings. So, please confirm what the correct term is and only use this one through the manuscript.
2) Section 2.1: It is mentioned that is used the Lysogeny broth (LB). However the composition of LB can have 3 different formulations with 3 different NaCl concentrations. So please describe the LB formula composition or mentioned the company of the LB source and type of LB used (Luria, Lennox or Miller).
3) Section 2.2, Bacteriophages and Strains: The 10 phages and the 126 E.coli strains should be listed on a table with all the information available about them. This table can be included in supplementary files.
4) Page 10: Please define the abbreviation EOP.
Author Response
Comments and Suggestions for Authors
The manuscript of Dufour et al. validates a semi-automated method to assess bacteriophages potency. The method described shows good reproducibility and will be important for speeding up the process of bacteriophage screening, in order to identify good therapeutic candidates for each specific bacterial infection.
Revisions:
1) In the manuscript the term “semi-automated” and “automated” is used. However, these terms have different meanings. So, please confirm what the correct term is and only use this one through the manuscript.
R: We harmonized the terminology as suggested to keep the term “automated”.
2) Section 2.1: It is mentioned that is used the Lysogeny broth (LB). However the composition of LB can have 3 different formulations with 3 different NaCl concentrations. So please describe the LB formula composition or mentioned the company of the LB source and type of LB used (Luria, Lennox or Miller).
R: We apologize for omitting that the medium used is LB Lennox. We modified the text accordingly.
3) Section 2.2, Bacteriophages and Strains: The 10 phages and the 126 E.coli strains should be listed on a table with all the information available about them. This table can be included in supplementary files.
R: As suggested, we have added a supplementary table containing all information about phages and strains used during this work.
4) Page 10: Please define the abbreviation EOP.
R: EOP is now defined and its meaning is explained.
Reviewer 3 Report
Comments and Suggestions for Authors
Dear Editor
The manuscript “High Throughput Bacteriophages Testing with Potency 2 Determination: Validation of an Automated Pipetting 3 and Phages Drop-Off Method” described determination of phages drop off method using semi-automated method.
General Comments:
1. There are multiple English errors, or the sentences are very difficult to understand i.e. Lines 17, 43-45 etc. It is suggested that the manuscript should be proof-read by some native English speaker.
2. The Authors have not included “quantitative results in the Abstract”.
3. The Introduction section needs to be modified to construct a strong background and to create interest in the scientific community regarding the current innovative study. The authors have described a single sentence “Line 43-45” to introduce the need of automated methodology.
Specific Comments:
1. Line 18: this is not necessary to add (agar plate), (PFU), Line 21-22: (phage titer, efficiency of 21 plaquing), (separately and assembled into one cocktail) and Line:
2. The methods section is well described, however Line 115-119, it is suggested that “table can be added to describe or summarize the characteristics of 126 E. coli isolates, as mentioned in Line 139 (E. coli-536).
3. Line 281-294: The paragraph described well the critical thinking of the authors. However, it is suggested that authors should cite side-by-side some of the references to create interests.
4. There is no Table 1 and Table 2, however it is cited in the text.
Thanks and Regards
Comments on the Quality of English LanguageEnglish Language requires extensive corrections.
Author Response
Dear Editor
The manuscript “High Throughput Bacteriophages Testing with Potency 2 Determination: Validation of an Automated Pipetting 3 and Phages Drop-Off Method” described determination of phages drop off method using semi-automated method.
General Comments:
- There are multiple English errors, or the sentences are very difficult to understand i.e. Lines 17, 43-45 etc. It is suggested that the manuscript should be proof-read by some native English speaker.
R: We have revised the manuscript in order to replace “frenchglish” by correct English language.
- The Authors have not included “quantitative results in the Abstract”.
R: We have revised the abstract accordingly.
- The Introduction section needs to be modified to construct a strong background and to create interest in the scientific community regarding the current innovative study. The authors have described a single sentence “Line 43-45” to introduce the need of automated methodology.
R: We have now added a new paragraph in the Introduction section that allows the reader to be more aware of the stakes related to automation and the future role of clinical laboratories in phage susceptibility testing.
Specific Comments:
- Line 18: this is not necessary to add (agar plate), (PFU), Line 21-22: (phage titer, efficiency of 21 plaquing), (separately and assembled into one cocktail) and Line:
R: We have removed unnecessary terms as suggested as well as unnecessary brackets. In order to ensure a precise definition of phage potency determination, which is not yet formally defined by regulatory agencies, we maintained the terms “phage titer and efficiency of plaquing”. We also kept the mention of the phage cocktail as it describes the work reported.
- The methods section is well described, however Line 115-119, it is suggested that “table can be added to describe or summarize the characteristics of 126 E. coli isolates, as mentioned in Line 139 (E. coli-536).
R: With the revised manuscript, we are submitting an additional table as supplementary material in which phages and strains are reported.
- Line 281-294: The paragraph described well the critical thinking of the authors. However, it is suggested that authors should cite side-by-side some of the references to create interests.
R: We thank the reviewer for this suggestion. We modified the text and added some references.
- There is no Table 1 and Table 2, however it is cited in the text.
R: We apologize for this inconvenience. It seems that the electronic file generated from our original submission, which included the two tables, has been corrupted making Table 1 and 2 unavailable for review. When we realized this issue we immediately informed the journal editing system by sending an updated file, which unfortunately was not transfer to this reviewer. When submitting the revision we will double/triple check that the electronic file includes all the necessary documents.
Round 2
Reviewer 3 Report
Comments and Suggestions for Authors
Dear Editor
The authors have significantly updated the manuscript, and it could be accepted in its current form. Thanks and Regards